

# Revision and validation of the Chinese version of the interpersonal reactivity index for couples for expectant couples

Juju Huang[1,2,*], Tengfei Liang[1,*], Jinzhi Li[1], Qiankun Liu[1], Jiaxue Pang[1], Yang Xu[1] and Hui Xie[1]

[1] College of Nursing, Bengbu Medical University, Bengbu, Anhui, China
[2] Department of Psychosomatic Medicine, Jieshou People's Hospital, Fuyang, Anhui, China
* These authors contributed equally to this work.

## ABSTRACT

**Objective:** This research seeks to evaluate the validity and reliability of the Interpersonal Reactivity Index for Couples (IRIC) to ensure it is culturally relevant to China, while also assessing its reliability and validity among a sample of pregnant women and their spouses.

**Methods:** A total of 402 couples were recruited from two hospitals in Anhui Province. The English version of the IRIC was translated into Chinese in accordance with Brislin's principles of cross-cultural translation. The reliability of the translated scale was assessed using Cronbach's α coefficient, split-half reliability, and test-retest reliability. The structural validity of the scale was examined through exploratory factor analysis (EFA) and confirmatory factor analysis (CFA). The Perceived Partner Responsiveness Scale served as a criterion measure to evaluate its correlation with the IRIC. All data analyses were performed using SPSS 26.0 and Mplus 8.3.

**Results:** The Chinese version of the IRIC comprises two dimensions and thirteen items (seven items pertaining to empathic concern and six items related to Perspective Taking). In the sample of pregnant women, the Cronbach's α coefficient for the Chinese version of the IRIC was 0.922, with coefficients of 0.871 for Empathic Concern and 0.909 for the Perspective Taking. The split-half reliability was 0.902 and the overall test-retest reliability of the scale was 0.996. In the sample of partners of pregnant women, the Cronbach's α coefficient was 0.938, with coefficients of 0.895 for empathic concern and 0.925 for Perspective Taking, and a split-half reliability of 0.898, while the overall test-retest reliability of the scale was 0.997. The content validity index at the scale level was 0.967, and at the item level, it ranged from 0.857 to 1.000. In the sample of pregnant women, the confirmatory factor analysis results indicated that the fit indices for the bi-factor model were satisfactory (Chi-square/degrees of freedom ($\chi^2$/df) = 1.331, comparative fit index (CFI) = 0.993, Tucker-Lewis index (TLI) = 0.987, normal fit index (NFI) = 0.972, goodness of fit index (GFI) = 0.958, root mean square error of approximation (RMSEA) = 0.041, standardized root mean square residual (SRMR) = 0.038). In the sample of partners, the confirmatory factor analysis results also demonstrated satisfactory fit indices for the bi-factor model ($\chi^2$/df = 1.588, CFI = 0.989, TLI = 0.976, NFI = 0.971, GFI = 0.961, RMSEA = 0.054, SRMR = 0.039). The scale successfully passed the equivalence test, with indices fitting well.

Corresponding author
Hui Xie, hui2122@sina.com

**Conclusion:** The findings suggest that the Chinese version of the IRIC exhibits strong reliability and validity, rendering it an effective instrument for evaluating the level of empathy between pregnant women and their partners. The translated scale also facilitates the early detection of couple empathy, providing a scientific foundation for the development of early personalized intervention strategies. Overall, this scale possesses clinical relevance and practical importance in enhancing marital satisfaction. However, future research should encompass a larger and more diverse population.

## INTRODUCTION

In recent years, the deterioration of marital relationships in China has gained increasing prominence (*Li et al., 2022*). According to the National Bureau of Statistics of China, the "Chinese Statistical Yearbook 2024" reports that in 2023, the national divorce registration reached 360.53 thousand couples, with a divorce rate of 2.56 per thousand. This marks the first time in the implementation of the "divorce moratorium" policy since 2021 that the national divorce rate exceeded 2.1 per thousand. This escalating divorce rate serves as a direct indicator of the diminishing stability of marital relationships, which has profound implications for individual mental health, family well-being, social structure, and economic development (*Ogihara, 2023*; *Zhai et al., 2024*).

In this context, emotional factors have become increasingly pivotal in marital and family relationships, as modern marriages now place a greater emphasis on the emotional bond between spouses (*Hou, Jia & Fang, 2024*). The decline in marital satisfaction has emerged as a primary driver of the rising divorce rate. Drawing from family systems theory, it is postulated that the interactions and relationships among family members exert a significant influence on individual psychology and behavior (*Brown, 1999*). Specifically, within marital relationships, the willingness and ability of spouses to engage in open and empathetic communication have a direct impact on marital satisfaction (*Girma Shifaw, 2024*).

In psychology, empathy refers to an individual's capacity to comprehend and share in the emotions of others (*Cuff et al., 2016*). General empathy is broadly defined as the inclination to empathize across various social contexts rather than being confined to specific relationships. In contrast, empathy in romantic relationships is referred to as empathy between partners (*Long, 1990*). Within the realm of romantic relationships, individuals who possess general empathy may find it more challenging to empathize with their partners in comparison to broader social contexts (*Birchler, Weiss & Vincent, 1975*). Consequently, a couple's empathy is better suited for analyzing interactions between partners within a dyadic relational framework, as well as the impact of empathy-based interactions on the emotional states of both partners (*Pistrang, Picciotto & Barker, 2001*). The ability to empathize between spouses is intricately linked not only to the quality of

their communication but also to marital satisfaction, conflict resolution skills, and the stability of the marriage (*Puryanto & Purwantiningsih, 2024*).

In marital relationships, open and honest communication facilitates the discovery of shared solutions when conflicts arise (*Ünal & Akgün, 2022*). Research has demonstrated a positive correlation between spouses' empathy and marital satisfaction. Couples with high levels of empathy tend to report greater satisfaction in their marital lives, experiencing heightened emotional connections and a sense of security (*Dong, Dong & Chen, 2022*). Conversely, a lack of empathy can lead to poor communication, frequent misunderstandings, and conflicts, thereby jeopardizing the stability and satisfaction of the marital relationship (*Wang & Zhao, 2023*). Individuals lacking empathy may struggle to provide adequate emotional support to their partners, resulting in emotional detachment and feelings of loneliness, which can adversely affect the quality and longevity of the marriage (*Cramer, 2004*; *Wang & Zhao, 2023*). During conflicts, a deficiency in empathy may prompt more selfish or aggressive behaviors, impeding problem resolution and relationship repair (*Puryanto & Purwantiningsih, 2024*). Therefore, it is evident that empathy, as a fundamental component of social support, is crucial for enhancing intimacy satisfaction and maintaining marital relationships (*Cramer, 2004*).

Empathy plays a pivotal role in marital relationships, particularly in the interactions between pregnant women and their spouses (*Zhu et al., 2024*). This capability profoundly facilitates a thorough understanding of each other's feelings and needs, especially during the unique phase of pregnancy, when pregnant women undergo drastic physical and emotional fluctuations. Empathy enables spouses to perceive these changes more acutely and, consequently, provide timely and appropriate support and comfort (*Zhu et al., 2024*). Furthermore, when confronted with the numerous uncertainties and pressures brought about by pregnancy, empathy becomes a core element for both spouses to jointly face challenges and collaborate in solving problems, effectively enhancing the stability and happiness of their marriage (*Spargo & Woodin, 2024*).

While various scales have been developed to assess empathy, they often lack specificity and are mostly general-purpose instruments. The Basic Empathy Scale (BES) is designed to measure an individual's empathy in general social interactions (*Jolliffe & Farrington, 2006*). The Jefferson Scale of Empathy—Healthcare Provider version (JSE-HP) focuses on the healthcare setting, aiming to facilitate understanding of patients' inner experiences (*Hojat et al., 2001*). However, there is currently no specific tool designed from a dyadic perspective to measure couple empathy. This study aimed to evaluate the validity and reliability of an instrument specifically developed to measure couple empathy in the Chinese context.

In 1980, Davis introduced the Interpersonal Reactivity Index (IRI) to assess general cognitive and emotional empathy (*Davis, 1983*). As research on intimate relationships evolved, Peloquin refined and validated the IRI in 2010, leading to the creation of the Interpersonal Reactivity Index for Couples (IRIC) (*Péloquin, Lafontaine & Brassard, 2011*). This scale has been extensively validated through cross-cultural studies, including samples from couples in Portugal (*Coutinho et al., 2016*), Chile (*Guzmán González et al., 2014*), Iran (*Ramezani et al., 2020*), and Poland (*Kaźmierczak & Karasiewicz, 2021*). However, it is important to note that the current IRIC has not been adequately validated

within pregnant populations. In particular, within the unique cultural context of China, characterized by distinct social customs, family values, and communication styles, the structure and items of the IRIC may necessitate further modification and optimization to address localized requirements. Consequently, there is an urgent need to develop and validate a couples' empathy measurement tool specifically designed for the Chinese population. Once the IRIC that is appropriate for the Chinese demographic is developed and successfully validated, it will empower healthcare professionals to more accurately evaluate and address empathy-related issues between pregnant women and their spouses in clinical settings, thereby significantly enhancing the specificity and effectiveness of clinical services. Additionally, the findings from this research are anticipated to provide a robust scientific foundation for government and relevant agencies in crafting family care and mental health policies, thus fostering the harmonious development of family relationships. By gaining a deeper insight into the current state and actual needs of couples' empathy levels, we can allocate social resources more effectively, thereby broadening the reach of family care and mental health services and improving their quality. Furthermore, by continuously refining and localizing the IRIC, we can better align it with the characteristics of Chinese culture, ultimately promoting the overall enhancement of family harmony and social well-being.

# MATERIALS AND METHODS

## Research design and participants

Between April and October 2024, a convenience sampling approach was utilized to conduct a survey involving 402 couples undergoing prenatal examinations at two medical facilities. The formal subjects for the survey were chosen through convenience sampling in accordance with the cross-cultural adaptation guidelines (*Kumar et al., 2021*). The sample size was established to be at least ten times the number of items on the scale; the Chinese version of the IRIC comprises 13 items. Taking into account a 20% attrition rate, the projected sample size was set at a minimum of 156 cases. During the recruitment phase, participant responses surpassed expectations, ultimately including 804 individuals, which consisted of 402 couples. This figure satisfied the predefined sample size criteria. Out of 862 individuals approached, 804 completed the Chinese version of the IRIC, yielding a response rate of 93.27%. Sample 1 included 404 individuals, comprising 202 pregnant women and 202 spouses as independent data, while Sample 2 included 400 individuals, comprising 200 pregnant women and 200 spouses as independent data. Although the couples were randomly assigned to the samples, all analyses treated the pregnant women and their spouses as independent samples. Exploratory Factor Analysis (EFA) was separately conducted in parallel for the pregnant sample ($n = 202$) and the spouse sample ($n = 202$). Confirmatory factor analysis (CFA) was independently conducted in the pregnant women ($n = 200$) and the spouses ($n = 200$) to validate the factor structure. Throughout the survey, participants received standardized instructions that clearly articulated the study's content and objectives. All participants were made aware that their involvement was voluntary, their responses would be kept confidential, and they had the option to withdraw at any time. The survey duration was approximately 20–30 min.

During data collection, pregnant women and spouses filled out questionnaires independently to ensure that both responses were not influenced by the other's emotions or opinions. From Sample 2, 30 couples were randomly selected to complete an online questionnaire two weeks after the initial survey to evaluate test-retest reliability, with all responses deemed valid and a 100% effective return rate for the questionnaire.

Inclusion criteria for the pregnant women: (1) Being married; (2) age ≥18 years; (3) gestational age ≥28 weeks; (4) possessing normal language communication skills and cognitive functions. Exclusion criteria for the pregnant women: (1) Having a confirmed diagnosis of psychological disorders such as depression or post-traumatic stress disorder. Inclusion criteria for the spouses: (1) Being married; (2) age ≥18 years; (3) possessing normal language communication skills and cognitive functions. Exclusion criteria for the spouses: (1) Having a confirmed diagnosis of psychological disorders such as depression or post-traumatic stress disorder. The diagnosis and exclusion of mental health disorders in this study were based on strict clinical criteria and a systematic data collection protocol. Specifically, medical records of pregnant women and their spouses were obtained through the health information management systems of local hospitals. Throughout the data collection process, participants with pre-existing mental health disorders were systematically excluded using predefined screening protocols. Prior to the investigation, written informed consent was obtained from all participants. This study has been approved by the Ethics Committee of Bengbu Medical University (Ethical Approval Number [2023] 254).

## Measures

The study employed a general information survey form along with a translated version of the 13-item IRIC. The Perceived Partner Responsiveness Scale was utilized for validation purposes.

## IRIC

The IRIC is a reliable and valid instrument for measuring the level of empathy between couples. It is composed of 13 items organized into two dimensions: Empathic Concern, which includes seven items, and Perspective Taking, which consists of six items. Each item is rated on a five-point Likert scale, ranging from 0 (does not apply) to 4 (applies very much). A higher total score on the IRIC indicates a higher level of empathy between couples.

## Translation and adaptation

To obtain the translation and usage permission of the IRIC, we contacted the original authors *via* email. Upon receiving authorization, we adhered to Brislin's (*Jones et al., 2001*) translation guidelines to adapt the IRIC into Chinese. The translation process involved several steps: Initially, two professionals with different backgrounds, including one with overseas study experience and a researcher in the field with high English proficiency, independently translated the scale from English to Chinese, producing two preliminary Chinese versions. Subsequently, the research team carefully compared, merged, and

**Table 1 Basic information of experts.**

| Number | Age | Years of work experience | Educational background | Professional title | Research area | Employing unit |
|---|---|---|---|---|---|---|
| 1 | 52 | 28 | PhD | Professor | Adult nursing | Institution of higher learning |
| 2 | 38 | 15 | PhD | Associate professor | Nursing psychology/Psychological counseling | Institution of higher learning |
| 3 | 57 | 35 | PhD | Professor | Adult nursing | Institution of higher learning |
| 4 | 48 | 25 | PhD | Associate professor | Nursing psychology/Psychological counseling | Institution of higher learning |
| 5 | 46 | 20 | PhD | Associate professor | Adult nursing/Nursing psychology | Institution of higher learning |
| 6 | 52 | 33 | Master | Professor | Nursing psychology | Institution of higher learning |
| 7 | 62 | 42 | Master | Professor | Adult nursing | Institution of higher learning |

discussed these two drafts to form a revised Chinese translation. Following this, we invited two additional professionals who had not previously encountered the original scale, also comprising a member with overseas study experience and a researcher in the field with high English proficiency, to independently back-translate the revised Chinese version into English. Further discussions and adjustments were made by the research team, leading to the development of the first draft of the Chinese-version IRIC. Seven experts were then invited to further modify and adjust the scale in light of Chinese cultural contexts, resulting in the second draft of the IRIC. Expert information is presented in Table 1. Next, using a convenience sampling method, 15 pregnant women and their spouses each from a tertiary hospital in Bengbu City were recruited to participate in a pilot survey using the second draft of the Chinese-version IRIC. The assessment focused on the appropriateness of the questionnaire completion time and gathered feedback from participants regarding any questions or suggestions about the scale content. Based on this feedback, further modifications were made to the questionnaire, culminating in the final version of the IRIC.

## Perceived partner responsiveness scale

The Perceived Partner Responsiveness Scale (*Yang et al., 2019*) is an instrument designed to measure an individual's perception of their partner's responsiveness within intimate relationships. In this study, the Chinese version translated by Yang Shucheng and colleagues was utilized as a criterion tool. The scale is composed of 12 items, and a seven-point Likert scale ranging from 1 (completely disagree) to 7 (completely agree) is employed for scoring. Participants are required to select a response that best reflects their perception. The mean score of all items is calculated, with higher scores indicating a greater perception of partner responsiveness by the individual.

## Statistical analysis

Data entry was conducted independently by two individuals using EpiData 3.1 software, and statistical analyses were performed using SPSS 26.0 and Mplus 8.3 software. Descriptive statistics were presented using frequencies, percentages, and means ± standard deviations. Item analysis was conducted using the critical ratio method and item-total correlation analysis. Pearson correlation analysis was employed to examine the product-moment correlation coefficient between each item and the total scale score, determining the homogeneity of the items and the scale.

The content validity index (CVI) of the scale was assessed using the scale-level content validity index (S-CVI) and item-level content validity index (I-CVI). An I-CVI ≥ 0.78 and S-CVI/Ave ≥ 0.90 indicated good content validity of the scale (*Polit, Beck & Owen, 2007*). The reliability of the scale was evaluated through internal consistency using Cronbach's α coefficient and split-half reliability. A Cronbach's α coefficient ≥ 0.70 suggested acceptable internal consistency of the scale. A split-half reliability > 0.70 indicated good split-half reliability (*Adamson & Prion, 2013*; *Parsons, 2021*). Test-retest reliability was assessed using the intraclass correlation coefficient (ICC), with an ICC > 0.75 indicating good reliability (*Koo & Li, 2016*).

The structural validity of the scale was tested using EFA. Sample 1 (*n* = 404) was used for item analysis and EFA. The suitability of the data for factor analysis was assessed using the Kaiser-Meyer-Olkin (KMO) measure and Bartlett's test of sphericity, with a KMO value > 0.8 and a significant Bartlett's test result indicating suitability for factor analysis. Sample 2 (*n* = 400) was used for CFA and criterion-related validity testing. The combined sample of Sample 1 and Sample 2 formed the total sample (*n* = 804) for reliability analysis and cross-gender measurement equivalence testing. Multi-group CFA was employed to test the cross-gender equivalence of the IRIC (*Cheung & Rensvold, 2002*). The Kolmogorov-Smirnov (K-S) goodness-of-fit test was conducted for each item, revealing that the data for each item were not normally distributed (*P* < 0.001). Therefore, robust maximum likelihood estimation was used for analysis.

## RESULTS

### Demographic results

Characteristics of the subjects are described in Table 2.

### Cultural adaptation

The Chinese version of the tool underwent meticulous revision by seven experts in the field of nursing to ensure the validity of its content. Additionally, the back-translated version was endorsed by the original tool's authors as accurate, thereby affirming the scientific rigor and meticulousness of the adaptation process. To more accurately capture and reflect the expressions of empathy among Chinese couples, the research team made necessary adjustments and optimizations to some phrases in the original version. For example, Item 2 was revised from "Sometimes I don't feel very sorry for my partner when he/she is having problems." to "When she is in trouble, I feel worried for him/her." Item 7 was changed from "If I'm sure I'm right about something, I don't waste much time listening to my

**Table 2 Demographic characteristics of pregnant women and their spouses ($n = 804$).**

| Characteristics | Sample 1 ($n = 404$) | | Sample 2 (400) | |
| --- | --- | --- | --- | --- |
| | Pregnant women ($n = 202$) | Spouses ($n = 202$) | Pregnant women ($n = 200$) | Spouses ($n = 200$) |
| Age, years ($n$, %) | | | | |
| 20–24 | 21 (10.4) | 11 (5.4) | 39 (19.5) | 22 (11.0) |
| 25–29 | 84 (41.6) | 79 (39.1) | 70 (35) | 78 (39.0) |
| 30–35 | 86 (42.6) | 85 (42.1) | 77 (38.5) | 80 (40.0) |
| >35 | 11 (5.4) | 27 (13.4) | 14 (7.0) | 20 (10.0) |
| Educational level ($n$, %) | | | | |
| Junior high school and below | 50 (24.7) | 46 (22.8) | 50 (25.0) | 45 (22.5) |
| High school and associate degree | 82 (40.6) | 91 (45.0) | 77 (38.5) | 83 (41.5) |
| Bachelor's degree and above | 70 (34.7) | 65 (32.2) | 73 (36.5) | 72 (36.0) |
| Job ($n$, %) | | | | |
| Employee | 65 (32.2) | 122 (60.4) | 56 (28.0) | 131 (65.5) |
| Maternity leave (Paternity leave) | 48 (23.8) | 54 (26.7) | 39 (19.5) | 36 (18.0) |
| Unemployed | 89 (44.0) | 26 (12.9) | 105 (52.5) | 33 (16.5) |
| Gender expectations for the baby ($n$, %) | | | | |
| Yes | 40 (19.8) | 42 (20.8) | 23 (11.5) | 34 (17.0) |
| Average | 46 (22.8) | 40 (19.8) | 47 (23.5) | 40 (20.0) |
| No | 116 (57.4) | 120 (59.4) | 130 (65) | 126 (63.0) |
| Sleep ($n$, %) | | | | |
| Very good | 33 (16.3) | 59 (29.2) | 23 (11.5) | 55 (27.5) |
| Good | 38 (18.8) | 62 (30.7) | 36 (18.0) | 65 (32.5) |
| Average | 114 (56.4) | 71 (35.1) | 129 (64.5) | 75 (37.5) |
| Poor | 17 (8.5) | 10 (5.0) | 12 (6.0) | 5 (2.5) |
| Usage of pregnancy-related apps ($n$, %) | | | | |
| Yes | 169 (83.7) | 110 (54.5) | 160 (80.0) | 107 (53.5) |
| No | 33 (16.3) | 92 (45.5) | 40 (20.0) | 93 (46.5) |
| Men's paternity experience ($n$, %) | | | | |
| Yes | – | 56 (27.7) | | 23 (11.5) |
| No | – | 146 (72.3) | | 177 (88.5) |
| Adverse obstetric history ($n$, %) | | | | |
| Yes | 29 (14.4) | – | 26 (13.0) | – |
| No | 173 (85.6) | – | 174 (87.0) | – |

partner's arguments" to "Even when I am sure I am right, I am still willing to take time to listen to his/her thoughts." Item 8 was altered from "When I see my partner being treated unfairly, I sometimes don't feel very much pity for him/her" to "When she is treated unfairly, I feel angry on his/her behalf." Item 10 was modified from ". In my relationship, I believe that there are two sides to every question and try to look at them both." to "I acknowledge that problems between us have two sides and can view them dialectically."

**Table 3 Cronbach's α coefficient of the revised 13-item Chinese version of the IRIC.**

| Index | Pregnant women | | | Spouses | | |
|---|---|---|---|---|---|---|
| | IRIC | Empathic concern | Perspective taking | IRIC | Empathic concern | Perspective taking |
| Reliability | 0.922 | 0.871 | 0.909 | 0.938 | 0.895 | 0.925 |
| Split-half reliability | 0.902 | 0.87 | 0.888 | 0.898 | 0.847 | 0.900 |
| Test-retest reliability | 0.996 | 0.859 | 0.923 | 0.997 | 0.798 | 0.874 |

## Item analysis

To assess the discriminatory power, the scores of the Chinese-version IRIC were sorted for both the pregnant women sample and the spouse sample. The top 27% of scores were categorized as the high-score group, while the bottom 27% were categorized as the low-score group. An independent samples t-test was performed on the data from these two groups. The results revealed statistically significant differences (t(110) = −30.761, 95% CI [−24.675 to −21.688], $P < 0.001$) in the pregnant women sample and (t(110) = −26.311, 95% CI [−21.979 to −18.900], $P < 0.001$) in the spouse sample when comparing item scores between the high-score and low-score groups. This indicates that each item in both versions of the scale possesses good discriminatory power.

## Content validity index

The Chinese-version IRIC exhibited an I-CVI ranging from 0.857 to 1.000 and a S-CVI of 0.967.

## Correlation between IRIC and its items

In both the pregnant women and their spouses samples, the correlation coefficients between the items of the Chinese-version IRIC and the total scale score ranged from 0.546 to 0.800 and from 0.648 to 0.823, respectively (all $P < 0.001$). These findings indicate that the items in both versions of the scale exhibit high homogeneity.

## Reliability analysis

See Table 3 for details.

## Validity analysis
### *Exploratory factor analysis*

The Chinese-version IRIC demonstrated suitable KMO values of 0.896 and 0.911 in the samples of pregnant women and their spouses, respectively. The Bartlett's test of sphericity yielded statistics of 1,716.43 ($P < 0.001$) and 2,006.472 ($P < 0.001$), respectively, indicating that the scale was appropriate for factor analysis in both sample groups. EFA extracted two common factors, with cumulative variance contributions of 64.569% and 68.47% in the samples of pregnant women and their spouses, respectively. Based on the EFA results and expert consultation, no items were added or deleted. Principal component analysis was used to extract two common factors, followed by varimax rotation for factor loading analysis. The results are presented in Table 4. Notably, Item 9, "I am often moved by little

**Table 4 Rotated component matrix.**

| Items | Factor (pregnant women) | | Factor (spouses) | |
|---|---|---|---|---|
| | Empathic concern | Perspective taking | Empathic concern | Perspective taking |
| IRIC1 | 0.801 | | 0.707 | |
| IRIC2 | 0.843 | | 0.745 | |
| IRIC3 | | 0.586 | | 0.694 |
| IRIC4 | 0.742 | | 0.832 | |
| IRIC5 | | 0.714 | | 0.82 |
| IRIC6 | 0.739 | | 0.852 | |
| IRIC7 | | 0.823 | | 0.829 |
| IRIC8 | 0.823 | | 0.738 | |
| IRIC9 | | 0.621 | | 0.738 |
| IRIC10 | | 0.745 | | 0.724 |
| IRIC11 | 0.431 | | 0.658 | |
| IRIC12 | | 0.848 | | 0.825 |
| IRIC13 | | 0.823 | | 0.752 |

**Table 5 Evaluation of the goodness of fit of the confirmatory factor analysis.**

| Index | Pregnant women | Model fit judgement | Spouses | Model fit judgement | Standard and critical value |
|---|---|---|---|---|---|
| $\chi^2$/df | 1.331 | Yes | 1.588 | Yes | <3 |
| CFI | 0.993 | Yes | 0.989 | Yes | >0.9 |
| TLI | 0.987 | Yes | 0.976 | Yes | >0.9 |
| NFI | 0.972 | Yes | 0.971 | Yes | >0.9 |
| GFI | 0.958 | Yes | 0.961 | Yes | >0.9 |
| RMSEA | 0.041 | Good fit | 0.054 | Reasonable | <0.05 (Good fit) <0.08 (Reasonable) |
| SRMR | 0.038 | Yes | 0.039 | Yes | <0.05 |

**Note:**
$\chi^2$/df, Chi-square/degrees of freedom; CFI, comparative fit index; TLI, Tucker-Lewis index; NFI, normed fit index; GFI, goodness of fit index; RMSEA, root mean square error of approximation; SRMR, standardized root mean square residual.

things between us," was reassigned from the Empathic Concern dimension to the Perspective Taking dimension.

*Confirmatory factor analysis*

From Table 5, we could see that the results of the CFA for the maternal version in the pregnant women sample indicated that the bifactor model exhibited good fit indices, with values of $\chi^2$/df = 1.331, CFI = 0.993, TLI = 0.987, NFI = 0.972, GFI = 0.958, RMSEA = 0.041, and SRMR = 0.038. Similarly, the CFA results for the spouses sample also demonstrated that the bifactor model had good fit indices, with values of $\chi^2$/df = 1.588, CFI = 0.989, TLI = 0.976, NFI = 0.971, GFI = 0.961, RMSEA = 0.054, and SRMR = 0.039.

**Table 6 Cross-gender equivalence test of the interpersonal reactivity index for couples.**

| Model | $\chi^2$ | df | AIC | CFI | TFI | RMSEA (90% CI) | SRMR | ΔCFI | ΔRMSEA |
|---|---|---|---|---|---|---|---|---|---|
| Model 1 | 292.169 | 100 | 22,150.631 | 0.974 | 0.959 | [0.060–0.078] | 0.040 | | |
| Model 2 | 307.570 | 111 | 22,144.032 | 0.973 | 0.963 | [0.058–0.075] | 0.050 | −0.001 | −0.01 |
| Model 3 | 361.660 | 122 | 22,176.122 | 0.967 | 0.958 | [0.062–0.078] | 0.054 | −0.006 | −0.004 |
| Model 4 | 482.738 | 73 | 22,271.200 | 0.953 | 0.945 | [0.072–0.088] | 0.078 | −0.014 | 0.013 |

**Note:**
The recommended critical values are ΔCFI ≤ 0.01 and ΔRMSEA ≤ 0.015. AIC, Akaike information criterion.

### Testing for measurement equivalence

The results of the test for equivalence demonstrated that all four equivalence models measured by the Chinese version of the IRIC were tenable. Specifically, the items of the scale possessed the same units and reference points across different groups, and the latent variable scores estimated from the observed variables were unbiased. These findings are presented in Table 6. The measurement equivalence of the scale between couples was established, indicating that the scale had equivalent measurement significance across different gender groups in China.

### Criterion-related validity

In the sample of pregnant women, the correlation coefficient between the Chinese-version IRIC and the criterion scale was 0.653 ($P < 0.001$), with correlation coefficients of 0.618 and 0.602 for the Empathic Concern and Perspective Taking dimensions, respectively (both $P < 0.001$). In the sample of spouses, the correlation coefficient between the Chinese-version IRIC and the criterion scale was 0.596 ($P < 0.001$), with correlation coefficients of 0.548 and 0.535 for the Empathic Concern and Perspective Taking dimensions, respectively (both $P < 0.001$).

## DISCUSSION

Empathy between couples plays a pivotal role in marital happiness and family well-being (*Carasso & Segel-Karpas, 2024*; *Dong, Dong & Chen, 2022*). However, its influence varies significantly across different cultural contexts. Take Israel, for instance, where research reveals that for first-time expectant couples, the higher the father's level of coupled empathy, the higher his reported postpartum depressive symptoms score at 6 months, yet coupled empathy does not mediate the transmission of parental relationship stress (*Zamir et al., 2023*). In sharp contrast, a study conducted in Canada demonstrates that coupled empathy plays a significant role in alleviating the stress experienced by new parents during the transition to parenthood, particularly with men's empathy during pregnancy enhancing their partners' relationship satisfaction (*Spargo & Woodin, 2024*). In the Chinese cultural context, research indicates that women's perception of their spouses' empathy is positively correlated with their overall well-being (*Zhu et al., 2024*). Therefore, it is imperative to develop a Chinese version of the IRIC. To more accurately capture and reflect the expressions of couple empathy in China, the research team made necessary

adjustments and optimizations to some phrases in the original version. These adjustments aim to make the instrument more culturally relevant and aligned with communication habits in China, thereby enhancing its applicability and accuracy among the Chinese population.

The I-CVI ranged from 0.857 to 1.00, and the S-CVI for the entire scale was 0.967, indicating that the Chinese version of the instrument possesses good content validity. In samples of pregnant women and their spouses, the Cronbach's α coefficients were 0.922 and 0.938, respectively, reflecting the stability and reliability of the scale's measurements, which are above the threshold of 0.80. The Cronbach's alpha coefficients and split-half reliabilities for each dimension were all above 0.80, indicating that the Chinese-version IRIC demonstrates good internal consistency.

Factor analysis was conducted on the 13 items in this study to explore their underlying structural dimensions. The results revealed that the cumulative variance contribution rates of the two factors were 64.569% and 68.47% in samples of pregnant women and their spouses, respectively. This finding indicates that these two factors collectively explain a substantial portion of the variability in the variables. All items had factor loading values greater than 0.4, and no dual loading phenomena were observed, further confirming the satisfactory construct validity of the scale. The factor analysis grouped these items into two primary factors: Factor 1 was labeled as "Empathic Concern," and Factor 2 was labeled as "Perspective Taking." Notably, Item 9, "I am often moved by little things between us," was initially attributed to the Empathic Concern dimension but was later included in the Perspective Taking dimension after further analysis. This adjustment differs slightly from the original version of the scale and may reflect the uniqueness of couple empathy expression in different cultural contexts. Additionally, the measurement equivalence of the scale between couples was confirmed, indicating that the scale has equivalent measurement meaning across different gender groups in China. This result is crucial for ensuring the fairness and validity of the scale in the context of gender differences and provides a solid psychometric foundation for cross-gender comparisons.

Specifically designed for couple relationships, the IRIC can more accurately capture and assess the unique empathy behaviors and experiences between spouses. This targeted approach makes the scale more sensitive and precise when assessing empathy in couple relationships. In clinical practice, the IRIC can serve as a tool for assessing couple relationships, particularly in situations involving marital issues and the impact of emotional disorders on marital relationships (*Kaźmierczak & Karasiewicz, 2021*; *Ramezani et al., 2020*). The Chinese-version of the IRIC has undergone reliability and validity testing, demonstrating its applicability in the Chinese cultural context. This indicates that the scale can effectively assess the level of empathy between spouses, thereby providing a basis for clinical interventions. In clinical applications, the IRIC can assist professionals in identifying potential issues within couple relationships and offering targeted intervention measures (*Long et al., 1999*; *Rosen, Mooney & Muise, 2017*).

Early screening and risk assessment for high-risk populations identification: Through a questionnaire, quickly screen families with low levels of spouse empathy, particularly those

expecting during pregnancy who face role adaptation pressure. For example, if a pregnant woman scores low in the "Perspective Taking" dimension, it may indicate difficulty in understanding emotional needs from the partner's perspective and requires priority intervention. Design personalized intervention modules. Based on the dimensions of the questionnaire, the intervention focus should be customized. Develop training modules such as role-playing and emotional diaries to enhance couples' ability to sense and feel emotions. Use exercises like "assuming you are the other, what support do you need now" to enhance mutual understanding. Develop digital intervention tools adapted to mobile devices. Embed the IRIC into a health app for pregnant couples, enabling real-time self-assessment and feedback. Utilize machine learning models to analyze questionnaire data and automatically generate personalized intervention recommendations. The Chinese version of the IRIC serves not only as an assessment tool but also as a bridge connecting assessment and intervention. By integrating it into couples' treatment plans, clinical workers can achieve a closed-loop management cycle of "assessment-intervention-assessment," providing a scientific pathway to enhance the mental health of pregnant couples and the quality of their marriages in China.

## Limitations

The current study's sample is limited to couples from Anhui Province's two hospitals, potentially restricting the representativeness and generalizability of the sample. Therefore, future research should validate the reliability and validity of the instrument in a broader couple population and compare coping patterns across different countries to refine the instrument's culturally sensitive items. Given the small sample size, increasing the sample size would provide more robust research results, particularly when employing statistical methods like factor analysis. Thus, increasing the sample size is an important direction for future research. Additionally, the study utilized a cross-sectional design, limiting the ability to explore changes in co-ping over time. Future research could conduct randomized controlled trials to assess the long-term effects of interventions guided by IRIC or adopt a longitudinal design to track changes in co-ping at different time points, thereby providing richer time-series data.

## CONCLUSION

Based on the findings of this study, the Chinese version of the IRIC demonstrates satisfactory reliability and validity, confirming its potential as a reliable tool for assessing the level of empathy between pregnant women and their spouses. This conclusion provides mental health professionals and researchers with an important instrument to more accurately measure and understand the dynamics of empathy within couples, especially during the special period of pregnancy. The application of the Chinese-version IRIC not only aids in assessing the current status of couple empathy but also provides a scientific basis for further interventions, potentially having a positive impact on enhancing marital happiness and family stability.

### Funding

This work was supported by the Anhui Provincial University Science Research Project in 2022 (Grant No. 2022AH040212). The funders had no role in study design, data collection and analysis, decision to publish, or preparation of the manuscript.

### Grant Disclosures

The following grant information was disclosed by the authors:
Anhui Provincial University Science Research Project in 2022: 2022AH040212.

### Competing Interests

The authors declare that they have no competing interests.

### Author Contributions

- Juju Huang conceived and designed the experiments, performed the experiments, analyzed the data, prepared figures and/or tables, authored or reviewed drafts of the article, and approved the final draft.
- Tengfei Liang analyzed the data, authored or reviewed drafts of the article, and approved the final draft.
- Jinzhi Li performed the experiments, analyzed the data, authored or reviewed drafts of the article, and approved the final draft.
- Qiankun Liu performed the experiments, prepared figures and/or tables, and approved the final draft.
- Jiaxue Pang performed the experiments, prepared figures and/or tables, authored or reviewed drafts of the article, and approved the final draft.
- Yang Xu performed the experiments, prepared figures and/or tables, and approved the final draft.
- Hui Xie conceived and designed the experiments, authored or reviewed drafts of the article, and approved the final draft.

### Human Ethics

The following information was supplied relating to ethical approvals (*i.e.*, approving body and any reference numbers):
Research Ethics Committee of Bengbu Medical University.

### Data Availability

The raw measurements are available in the Supplemental File.

### Supplemental Information

Supplemental information for this article can be found online at http://dx.doi.org/10.7717/peerj.19505#supplemental-information.

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
