# Peer review of "Revision and validation of the Chinese version of the interpersonal reactivity index for couples for expectant couples"

_PeerJ, doi:10.7717/peerj.19505_

## Round 0.1 · original submission · Major Revisions

Please respond to the comments from both reviewers.

Reviewer 1 ·

Basic reporting

The article is well-written in professional and clear English. The introduction provides a strong theoretical background, citing relevant and up-to-date studies. The study is supported by a well-curated selection of references, incorporating recent and relevant literature to provide a strong theoretical foundation. Figures and tables are well-prepared, relevant, and adequately described.
Weaknesses:
• The introduction is highly detailed and could be slightly condensed for better readability.
• Some sentences are repetitive and could be simplified (e.g. lines 48–56:58-64; lines 65-70:75-81)
Suggestions for improvement:
• Condense the introduction by removing repetitive elements and focusing on key points

Experimental design

The study design is solid and well-explained. The objective is clearly defined.
The study benefits from a sufficiently large sample size of 402 couples, ensuring robust statistical power. It employs a rigorous translation and adaptation methodology, adhering to established guidelines for cross-cultural research. Additionally, the analysis is strengthened by the use of advanced statistical techniques, enhancing the reliability and validity of the findings.
Weaknesses:
• The sample was collected using convenience sampling and was limited to two hospitals in one province, reducing generalizability.
• The study lacks longitudinal data, which would help assess the scale’s long-term stability.
• Cultural differences that might influence item interpretation are not thoroughly discussed.

Suggested improvements:
• Discuss cultural limitations more thoroughly and explore the possibility of further adaptations.
• Consider longitudinal studies in the future to test the scale’s long-term stability.

Validity of the findings

The results are presented clearly and supported by solid statistical analyses. The reliability and validity measures indicate strong internal consistency and discriminant capacity.

Suggested improvements:
• Expand the discussion on potential clinical applications and how the scale could be used in couple therapy programs.

Additional comments

No additional comments.

Reviewer 2 ·

Basic reporting

The manuscripts uses clear and unambiguous language throughout; the introduction section could be strengthened by addressing the following concerns:

Line 51 - please include what percentage of all marriage is that number, is it an increase from previous years? Adding that information would make a more compelling argument
APA style needs to be adjusted throughout the manuscript e.g. line 59 only cite first author followed by et al., also only report two numbers after the decimal point ect.

Line 68 - I think you mean empathy between partners not couples - correct this sentence

Starting with line 70 - that whole page/argument is confusing. Not sure if the explanation or comparison between general empathy and couples empathy is needed and if so, highlighting lack of adequate measures would be more helpful here (rather than in the discussion section). Also, not sure how the concept of positive resonance relates here.

Line 111 - be careful not to blame mothers for children's developmental concerns - that's the undertone of this statement. See if you can include more recent literature here and also comment on father's empathy and child outcomes.

Line 121- seems like a far fetched statement - the assessment by itself does not lead to that - please adjust the language here.

Experimental design

The research question is well-defined and meaningful; the investigation is preformed with high standard. I do have some concerns regarding the methods section. Can you clarify how the groups were established? Was sample 1 consisting only of pregnant women and sample 2 of fathers? If not, how can you ensure independence of observations? If this was not addressed that would be a concern for major revision.

Also how was mental health disorder diagnosed verified for exclusion? Please report if it was a self-report. For the item-analysis section - make sure you report t-test finings in APA style and include degrees of freedom.

Validity of the findings

The discussion section includes information that is better suited to the introduction and methods. I think it would be interesting to further discuss cultural differences in terms of showing empathy.
Line 367 - it is unclear how it outpreformed other scales - please provide more evidence or change this statement.

---

## Round 0.2 · Minor Revisions

One reviewer has some remaining concerns that should be addressed in the next revision.

Reviewer 2 ·

Basic reporting

I appreciate the changes authors implemented in the introduction section. I think it makes a more compelling argument.

Experimental design

i appreciate the additional information and clarification. However, I still believe that the methods section could be a bit more clear. What the authors add in their revision was independence of responses between spouses during data collection (sounds like that was something that was considered so that is great). However, what I was wondering about is the nonindepenence of responses when utilizing data from both spouses in one analysis. Looking at the raw data it looks like males and female data were analyzed separately so that would prevent that issue. I think what is confusing is the statement on page 8 "Group 1, which included 202 couples, was designated for exploratory factor analysis; Group 2, consisting of 200 couples, was allocated for confirmatory factor analysis" it suggests that the couples data was analyzed together, when later in the data analysis section pregnant women and spouses are described as separate samples. Including the composition of each group would make it more clear.

Validity of the findings

I appreciate the addition of cultural differences and more context.

---

## Round 0.3 · accepted · Accept

The reviewers' comments have been adequately addressed, and the manuscript is now ready for publication.

Reviewer 2 ·

Basic reporting

No additional comments

Experimental design

Thank you for making additional changes, I do not have any further comments.

Validity of the findings

No additional comments.

Additional comments

N/A